# No Impact of Smoking Status on Breast Cancer Tumor Infiltrating Lymphocytes, Response to Neoadjuvant Chemotherapy and Prognosis

**DOI:** 10.3390/cancers12102943

**Published:** 2020-10-12

**Authors:** Vanille Simon, Lucie Laot, Enora Laas, Sonia Rozette, Julien Guerin, Thomas Balezeau, Marion Nicolas, Jean-Yves Pierga, Florence Coussy, Marick Laé, Diane De Croze, Beatriz Grandal, Judith Abecassis, Elise Dumas, Florence Lerebours, Fabien Reyal, Anne-Sophie Hamy

**Affiliations:** 1Department of Surgical Oncology, Institut Curie, University Paris, 75005 Paris, France; vanille.simon@aphp.fr (V.S.); lucie.laot@aphp.fr (L.L.); enora.laas@curie.fr (E.L.); marion.nicolas@aphp.fr (M.N.); beatriz.grandalrejo@curie.fr (B.G.); 2Department of Medical Oncology, Institut Curie, University Paris, 75005 Paris, France; sonia.rozette@aphp.fr (S.R.); jean-yves.pierga@curie.fr (J.-Y.P.); florence.coussy@curie.fr (F.C.); florence.lerebours@curie.fr (F.L.); anne-sophie.hamy-petit@curie.fr (A.-S.H.); 3Data Factory, Data Office, Institut Curie, 25 rue d’Ulm, 75005 Paris, France; julien.guerin@curie.fr (J.G.); thomas.balezeau@curie.fr (T.B.); 4Department of Tumor Biology, Institut Curie, 75005 Paris, France; marick.lae@curie.fr (M.L.); diane.decroze@curie.fr (D.D.C.); 5Residual Tumor & Response to Treatment Laboratory, RT2Lab, Translational Research Department, INSERM, U932 Immunity and Cancer, University Paris, 75005 Paris, France; judith.abecassis@curie.fr (J.A.); elise.dumas@curie.fr (E.D.)

**Keywords:** smoking status, breast cancer, neoadjuvant chemotherapy, tumor infiltrating lymphocytes, pathologic complete response, prognosis

## Abstract

**Simple Summary:**

Tobacco use is associated with an increase in breast cancer mortality. Pathologic complete response (pCR) rate to neoadjuvant chemotherapy is influenced by tumor-infiltrating lymphocyte (TIL) levels and is associated with a better long-term survival outcome. The aim of this study was to evaluate the impact of smoking status on TIL levels, response to neoadjuvant chemotherapy and prognosis for breast cancer patients. We retrospectively assessed pre- and post-neoadjuvant chemotherapy tumor infiltrating lymphocyte (TILs) levels and pathological complete response (pCR) rates in a cohort of 956 specimens of breast cancer (BC) patients treated with neoadjuvant chemotherapy, according to their smoking status. To our knowledge, this is the largest cohort of BC patients used to study this topic so far. We found no impact of smoking status on tumor infiltrating lymphocyte levels, response to neoadjuvant chemotherapy and prognosis in the whole population and within each BC subtype.

**Abstract:**

Tobacco use is associated with an increase in breast cancer (BC) mortality. Pathologic complete response (pCR) rate to neoadjuvant chemotherapy (NAC) is influenced by tumor-infiltrating lymphocyte (TIL) levels and is associated with a better long-term survival outcome. The aim of our study is to evaluate the impact of smoking status on TIL levels, response to NAC and prognosis for BC patients. We retrospectively evaluated pre- and post-NAC stromal and intra tumoral TIL levels and pCR rates on a cohort of T1-T3NxM0 BC patients treated with NAC between 2002 and 2012 at Institut Curie. Smoking status (current, ever, never smokers) was collected in clinical records. We analyzed the association between smoking status, TIL levels, pCR rates and survival outcomes among the whole population, and according to BC subtype. Nine hundred and fifty-six BC patients with available smoking status information were included in our analysis (current smokers, *n* = 179 (18.7%); ever smokers, *n* = 154 (16.1%) and never smokers, *n* = 623 (65.2%)). Median pre-NAC TIL levels, pCR rates, or median post-NAC TIL levels were not significantly different according to smoking status, neither in the whole population, nor in any BC subtype group. With a median follow-up of 101.4 months, relapse-free survival (RFS) and overall survival (OS) were not significantly different by smoking status. We did not find any significant effect of tobacco use on pre- and post-NAC TILs nor response to NAC. Though our data seem reassuring, BC treatment should still be considered as a window of opportunity to offer BC patients accurate smoking cessation interventions.

## 1. Introduction

Neoadjuvant chemotherapy (NAC) is currently prescribed for patients with locally advanced breast cancers (BC) (T3–T4), triple negative (TNBC), HER2-positive status or positive nodal status. Beyond increasing breast-conserving surgery rates [1], it also serves as an in vivo chemosensitivity test and the analysis of residual tumor burden may help in understanding resistance to treatment [2]. Moreover, pathologic complete response (pCR) after NAC has been associated with a better long-term survival outcome [1,3].

Denkert et al. first showed that the amount of stromal immune infiltration was positively associated with pathological complete response (pCR) after NAC [4]. These results were recently confirmed in a pooled analysis of a large cohort of 3771 patients receiving NAC from the German Breast Group [5], showing that the relationship between tumor-infiltrating lymphocyte (TIL) levels and pCR translated into a disease-free survival advantage in HER2-positive and TNBC tumors. The drivers of immunosurveillance derive from both tumor-intrinsic characteristics, and extrinsic factors related to the host or the environment [6]. Among endogenous tumor characteristics, BC subtype and proliferative patterns are the main factors associated with immune infiltration. Extrinsic factors including, notably, environment (tobacco, alcohol), nutritional factors and diet have been studied less extensively. The identification of factors associated with changes in the microenvironment is of particular interest, as lifestyle factors are actionable and their changes could theoretically improve prognosis.

Tobacco smoking is known to be associated with an increased risk of several cancer types, including larynx, oropharynx, esophagus, lung, bladder, kidney, urinary tract, cervix, colorectum and gastrointestinal tract, and acute myeloid leukemia [7]. Tobacco deregulates many biological pathways, and induces inflammation, impaired immune function and DNA damage [8], leading to an increase in tumor proliferation, invasion and angiogenesis. Regarding BC, tobacco is associated with post-operative complications [9], radiation-induced toxicities [10], and altered quality of life [11]. A recent meta-analysis on 39,725 patients reported that smoking increased the risk of BC, all-cause and specific mortality [12]. Studies have demonstrated that BC patients who smoke have a higher risk of second primary cancer [13], and ipsilateral lung cancer [14] when combined with radiotherapy. However, there are few data on the relationship between cigarette smoking, immune infiltration and response to NAC in BC.

The aim of our study is to analyze the relationships between smoking status at BC diagnosis, tumor infiltrating lymphocytes, response to NAC and prognosis on a large cohort of BC patients treated with NAC.

## 2. Results

### 2.1. Baseline Patients’ and Tumors’ Characteristics

In the whole population (*n* = 956), the mean age was 48 years old at diagnosis, and the mean body mass index (BMI) was 24 kg/m^2^ (range 16–47). Most of BCs were in T2 stage (66.2%), grade III (61%), and 550 patients (57.6%) had positive axillary nodes. Patient’s repartition by subtype was as follows: luminal (*n* = 410, 42.9%), TNBC (*n* = 305, 31.9%), HER2-positive (*n* = 241; 25.2%) (Table 1).

At BC diagnosis, 179 patients were current smokers (18.7%), 154 were ever smokers (16.1%) and 623 women were never smokers (65.2%). Among 333 patients with a previous tobacco history (34.8%), 23% (*n* = 77) were heavy smokers > 20 pack-years. The median pack-year was 15 (range 1–100). Patients’ characteristics according to smoking status are described in Table 1. Current and ever smokers were statistically younger (46.4 versus 48.8 y.o., *p* = 0.002) and thinner (BMI 24 versus 25, *p* = 0.006) than never smokers. Conversely, no tumor characteristic was statistically different according to smoking status, among tumor size, tumor grade, histology, mitotic index, ER, PR, HER2 or nodal status (Table 1).

Pre-NAC str TIL levels were available for 632 patients and were not significantly different between current, ever, and never smokers (15%, 15% and 20%, respectively, *p* = 0.1, Figure 1). This was true in the whole population (Figure 1A) and in each BC subtype population (Figure 1B). Similarly, intra-tumoral (IT) TIL levels were not significantly different according to smoking status in the whole population (Figure 1C current: 5%, ever: 5% and never 5%, *p* = 0.26), neither were they in each BC subtype population (Figure 1D). Among current smokers, no difference was found in stromal or IT TIL levels according to smoking intensity (Appendix A).

### 2.2. Response to Treatment and Post-NAC Immune Infiltration

After NAC, the pCR rate was 25.8% (247/956) and this rate was different according to BC subtype (luminal: 6.8% (28/410); TNBC: 39.7% (121/305), HER2-positive: 40.7% (98/241), *p* < 0.001). pCR rates were not significantly different in current, ever or never smokers (20.7%, 29.2% and 26.5% respectively, *p* = 0.17, Table 2, Figure 2A). Similar results were found in each BC subtype population (luminal: *p* = 0.63; TNBC: *p* = 0.27; HER2-positive: *p* = 0.22, Figure 2B). In the same vein, no difference was seen regarding the residual cancer burden (RCB) index (Appendix A).

Post-NAC str and IT TILs were available for 632 and 429 patients, respectively. Post-NAC median str TIL levels were not significantly different between current, ever and never smokers in the whole population (10%, 7% and 10%, respectively, *p* = 0.108, Figure 3A) nor in each BC subtype (Figure 3B). Similar results were found for post-NAC IT TIL levels both in the whole population (5%, 5% and 5%, respectively, *p* = 0.254) and in each BC subtype population (Figure 3C,D respectively).

### 2.3. Survival Outcomes

With a median follow-up of 101.4 months, 293 patients experienced relapse, and 173 died. Tobacco smoking was not significantly associated with relapse-free survival (RFS), either in the whole population, or after stratification by BC subtype (Figure 4). Similar results were observed for overall survival, where tobacco smoking at diagnosis had no impact at all on mortality in the whole population (Appendix A). After subgroup analysis by BC subtype, patients with luminal BC who were current smokers had a worse overall survival when compared with ever or never smokers (Pinteraction smoking status /BC subtype = 0.11), but this association was no longer significant after multivariate analysis (Appendix A).

## 3. Discussion

In this retrospective study evaluating the association between smoking status and oncologic outcomes, we found no significant association between smoking status, pre- and post-NAC immune infiltration, response to treatment or prognosis in a large cohort of BC patients treated with NAC. These findings are of interest, because a significant increase in female tobacco consumption has been observed worldwide over the last century [15], and whether tobacco use could affect response to chemotherapy and prognosis has been barely explored so far.

To our knowledge, only one previous study evaluated the relationship between smoking and the immune microenvironment in BC. On a retrospective study assessing the relationship between tobacco use, TIL levels and pCR in 149 women with HER2-positive and TNBCs, Takada and colleagues [16] found that TIL levels and pCR rates were significantly higher in the high smokers group (defined by more than 2.5 pack-year) than in the low smokers group (TIL levels: 72.1% versus 56.6%, *p* = 0.043; pCR 62.8% versus 44.3%, *p* = 0.042). However, while using the binning comparing smokers versus no smokers, the difference in TILs density and pCR rates was no longer significant (*p* = 0.065 and 0.075, respectively). The latter results are consistent with the current results of our study, showing no association between smoking status and pre-NAC stromal immune infiltration on the one hand, and pCR rates on the other hand.

Conversely, tobacco use was reported to have a negative impact on survival in patients treated with endocrine therapy. In a cohort of 1116 patients, Persson et al. [17] showed an increased risk of recurrences (HR 2.97, 95%CI [1.81–9.72]), distant metastasis (HR 4.19; 95%CI [1.81–9.72]) and death (HR 3.52; 95%CI [1.59–7.81]) among aromatase-inhibitors treated patients who smoked compared with non-smokers. However, there was no effect in patients treated with tamoxifene.

In organs directly affected by tobacco, several studies [18,19,20,21,22] showed that perpetuation of tobacco use during radiotherapy was associated with a reduction in its effectiveness and overall survival. In head and neck cancers, Chen et al. [18] reported that active smokers during radiotherapy had a significantly inferior 5 year overall survival (23% vs. 55%, *p* < 0.05), locoregional control (58% vs. 69%, *p* < 0.005), and disease-free survival (42% vs. 65%, *p* < 0.05) when compared with the former smokers who had quit before radiation therapy. In lung cancers, Videtic et al. [22] showed a significantly better 5 year overall survival in weaned patients compared to active smokers during radiotherapy (8.9% versus 4% respectively, *p* = 0.0017).

A strong biological and preclinical rationale could have supported the working hypothesis that the immune response to tumors and sensitivity to chemotherapy may be impaired—or modified—by tobacco. Indeed, nicotine deregulates cell proliferation, apoptosis, migration, invasion, angiogenesis, inflammation, epithelial-mesenchymal transition and cell-mediated immunity in a wide variety of cells, leading to enhanced tumor growth and metastasis [8,23]. The effects of nicotine are usually mediated through the nicotinic acetylcholine receptors (nAChRs) [24,25,26], that in turn activate several oncogenic pathways as Ras/Raf/MAPK and PI3K/AKT cascades. Smoking has been reported to induce chemoresistance in vitro, in colorectal [27], bladder [18], pancreatic [28] and nasal [29] cancers. Nicotine suppressed chemotherapeutic-induced apoptosis of breast cancer cells in vitro [30], via the signaling cascade involving STAT3, galectine-3, and a nicotinic acetylcholine receptor. Tobacco use may also modify pharmacodynamics of anticancer agents [31,32]. In lung cancer, smokers receiving erlotinib or camptothecine showed a rapid clairance requiring a higher dose to reach an equivalent systemic effect than never smokers [33].

Recent analyses of cancer genomes have highlighted an association between mutational processes and the catalogue of somatic single nucleotide substitutions observed in a tumor sample. In particular, specific processes are characterized by preferential substitution types, and sequence context (defined by the two flanking nucleotides) [34].

In a systematic analysis of 5243 cancers of types for which tobacco smoking confers an elevated risk (breast cancer was not included), signature 4—the main marker of tobacco-smoking-caused mutations—was only identified in cancers from tissues directly exposed to tobacco smoke, suggesting that the increased risk associated with tobacco may be mediated by mechanisms other than an increased mutation load as previously believed [35]; immunity could be a good candidate, that could be further explored in other cancer types without direct exposition to smoke, such as pancreatic or bladder.

Tobacco may also play a role in inflammation and the deregulation of innate and adaptative immunity, and notably affects T-cell lymphocytes’ function [36]. In non-small-cell lung cancers, Deng et al. showed that current/ever smokers had higher PDCD1 and CTLA-4 expression in tumor tissues, compared with never smokers (PDCD1 median 142 vs. 36, *p* < 0.01; CTLA-4 median 152 vs. 59, *p* < 0.01) [37]. In non-small-cell lung cancer, both nivolumab and MPDL3280A [38] (a PD-L1 antibody) have been reported to be more active in current/ever smokers than in never-smokers.

However, despite such abundant scientific rationale, our study provides reassuring data on the impact of smoking on BC outcomes in the neoadjuvant setting.

Our study has several strengths. First, it included a large number of patients, with the availability of paired matched pre- and post-NAC samples for 632 patients, thus providing unprecedented data on post-NAC stromal and intra-tumoral TIL assessment.

Second, we used a very standard methodology for TIL assessment [39], making our study reproducible worldwide, since TIL reading has been incorporated in routine practice since 2015.

Third, the choice of a continuous TIL scoring was an asset of our study. Indeed, several cut-offs for TIL scoring have been studied without being formally validated, and it is well known that TIL levels depend on BC histological subtypes. Even while presenting extensive analyses with subtype subsetting, we failed to identify any significant association between smoking status and the clinical outcomes we studied.

It also has several limits, such as the lack of double assessment using digital pathology. However, we previously showed [40] that the results we obtained using pathologist manual assessment were comparable with those obtained by digital assessment in the Neo-t-ANGo trial and the ARTemis trial. Finally, we have not yet performed immunostaining to separate out the immune subpopulations, which will enable us to determine whether there is any enrichment in immunosuppressive signaling. However, there is currently no clear consensus as to which single antibody or antibody combination should be used, and their interpretation is not standardized. One advantage of quantitative TIL assessment is that it could be performed routinely in any pathology department with no real increase in technical costs. In addition, some studies have reported a positive correlation between the numbers of unstained TILs and CD8+ TILs [41], CD3 counts or counts for other immune subpopulations (CD3+, CD20+, CD68+) [42], supporting the notion that quantitative assessments could serve as a relevant surrogate marker. However, from a research standpoint, extensive characterization of the lymphocyte infiltrate remaining in residual tumors to determine the subsets of TILs present could be of interest.

Finally, taking into account the well-known benefits of weaning on wound healing, quality of life, and overall survival, BC treatment and follow-up should be considered as windows of opportunity to address tobacco use and to offer patients accurate smoking cessation alternatives.

## 4. Materials and Methods

### 4.1. Patients and Tumors

This was a retrospective observational study. We analyzed a cohort of 956 T1-3NxM0 patients with invasive BC (NEOREP Cohort) treated with NAC at Institut Curie between 2002 and 2012 with data available on smoking status (NEOREP Cohort, CNIL declaration number 1547270) [43]. We included patients with unilateral, non-recurrent, non-inflammatory, non-metastatic tumors for whom NAC was indicated. Every patient received NAC, followed by surgery, radiotherapy and endocrine therapy when indicated. The study was approved by the Breast Cancer Study Group of Institut Curie and was conducted according to institutional and ethical rules regarding research on tissue specimens and patients (the ethic code for our study is CNIL declaration number 1547270). Written informed consent from the patients was not required by French regulations.

### 4.2. Tobacco Smoking

Data regarding history of smoking was collected retrospectively in May 2018 in clinical records (either in oncology or gynecology first consultation, either in anesthesiology consultations) for the purpose of the current study. We defined current smokers as active smokers at the time of BC diagnosis, ever smokers as patients with a prior history of smoking having stopped before BC diagnosis, and never smokers as patients who had never smoked. We also documented smoking intensity through the use of pack-years, which is a measurement unit calculated by multiplying the number of packs of cigarettes smoked per day by the number of years the person has smoked. Heavy smokers were defined as 20 or more pack-years [44].

### 4.3. Treatments

Patients were treated according to national guidelines. NAC regimens changed over time (anthracycline-based regimen or sequential anthracycline-taxane regimen), and trastuzumab was used in an adjuvant and/or neoadjuvant setting since 2005 for HER2-positive breast cancer. Surgery was performed four to six weeks after the end of chemotherapy. Most patients (98.2%, *n* = 1127) received adjuvant radiotherapy. Endocrine therapy (tamoxifen, aromatase inhibitor, and/or GnRH agonists) was prescribed when indicated.

### 4.4. Tumor Samples

BC tumors were classified into subtypes (TNBC, HER2-positive, and luminal) on the basis of immunohistochemistry. Estrogen receptor (ER) and progesterone receptor (PR) status were determined as follows. Tissue sections were rehydrated, and antigen retrieval was carried out in citrate buffer (10 mM, pH 6.1). The sections were then incubated with antibodies against ER (clone 6F11, Novocastra, Leica Biosystems, Newcastle, UK; 1/200) and PR (clone 1A6, Novocastra, 1/200). Antibody binding was detected with Vectastain Elite ABC peroxidase-conjugated mouse IgG kit (Vector, Burlingame, CA, USA), with diaminobenzidine (Dako A/S, Glostrup, Denmark) as chromogen. Positive and negative controls were included in each run. According to French recommendations, cases were considered positive for ER and PR if at least 10% of tumor nuclei were stained [45]. Tumors were considered hormone receptor (HR)-positive when positive for either ER or PR (referred to hereafter as “luminal”), and HR-negative when negative for both ER and PR. HER2 expression was determined by immunohistochemistry using a monoclonal anti-HER2 antibody (CB11, Novocastra, Newcastle, UK; 1/800). Scoring was performed according to American Society of Clinical Oncology (ASCO)/College of American Pathologists (CAP) guidelines. Scores of 3+ were reported as positive, and scores of 0/1+ as negative. Tumors with scores of 2+ were tested by fluorescence in situ hybridization (FISH). FISH was performed using a HER2-gene-specific probe and a centromeric probe for chromosome 17 (PathVysion HER-2 DNA Probe kit, Vysis-Abbott, Abbott Park, IL, USA) according to manufacturers’ instructions. HER2 gene amplification was defined according to ASCO/CAP guidelines. An average of 40 tumor cells per sample were evaluated and mean HER2 signals per nuclei were calculated. A HER2/CEN17 ratio ≥ 2 was considered positive, and a ratio < 2 was considered negative [46].

BC subtypes were defined as follows: tumors positive for either ER or PR, and negative for HER2 were classified as luminal; tumors positive for HER2 were considered HER2-positive BC; tumors negative for ER, PR, and HER2 were considered as triple-negative BC (TNBC).

### 4.5. Pathological Review

Pretreatment core needle biopsy specimens and the corresponding post-NAC surgical specimens were reviewed independently by two experts in breast diseases. Formalin-fixed paraffin-embedded (FFPE) tumor tissue samples were studied. Pre- and post-NAC stromal (str) and intra-tumoral (IT) TILs were reviewed between January 2015 and March 2017 according to the recommendations of the international TILs Working Group [39]. Further details on the TILs review are available in [40].

Response to treatment was retrospectively reviewed and was assessed by: (i) pathological complete response, defined as the absence of residual invasive cancer in the breast and axillary nodes (ypT0/ ypN0) after neoadjuvant chemotherapy [47]; (ii) residual cancer burden (RCB) indices as described by Symmans [48], with the web-based calculator freely available via the Internet (www.mdanderson.org/breastcancer_RCB). pCR corresponded to an RCB 0 index. Further details on RCB review are available in ref [49].

### 4.6. Study Endpoints

The aims of the study were to analyze the association between smoking status at diagnosis and: (i) pre- and post-NAC str TILs; (ii) response to NAC assessed by pathological complete response (pCR) and RCB index; (iii) prognosis assessed by relapse-free survival (RFS). Relapse-free survival (RFS) was defined as the time from surgery to death, loco-regional recurrence or distant recurrence, whichever occurred first. Overall survival (OS) was defined as the time from surgery to death. For patients for whom none of these events were recorded, we censored data at the time of last known contact. Survival cutoff date analysis was 1 February 2019.

### 4.7. Statistical Analysis

The population was described in terms of frequencies for qualitative variables or medians and associated range for quantitative variables. Pre- and post-NAC TIL levels and RCB index were analyzed as continuous variables. All analyses were performed on the whole population and in each subgroup of BC subtype (luminal, TNBC, HER2-positive). TIL levels and qualitative variables in classes were compared with ANOVA or Mann–Whitney tests, when appropriate. Comparisons of proportion of samples were investigated by chi-squared and Fisher tests.

Factors predictive of pCR were introduced into a univariate logistic regression model. A multivariate logistic model was then implemented. Covariates selected for multivariate analysis were those with a *p*-value likelihood ratio test below 0.05 in univariate analysis. Survival was described using a Kaplan–Meier estimate and comparison between survival curves was performed with the Log-rank test. Estimation of hazard ratios (HR) and their associated 95% confidence interval (CI) was carried out using the Cox proportional hazard model. The significance threshold was 5%. Analyses were performed with R software, version 3.1.2 (R Foundation for Statistical Computing, Vienna, Austria, 2009, www.cran.r-project.org).

## 5. Conclusions

We did not find any significant effect of tobacco use on pre- and post-NAC TILs nor response to NAC. Though our data seem reassuring, BC treatment should still be considered as a window of opportunity to offer BC patients accurate smoking cessation interventions.

## Figures and Tables

**Figure 1 cancers-12-02943-f001:**
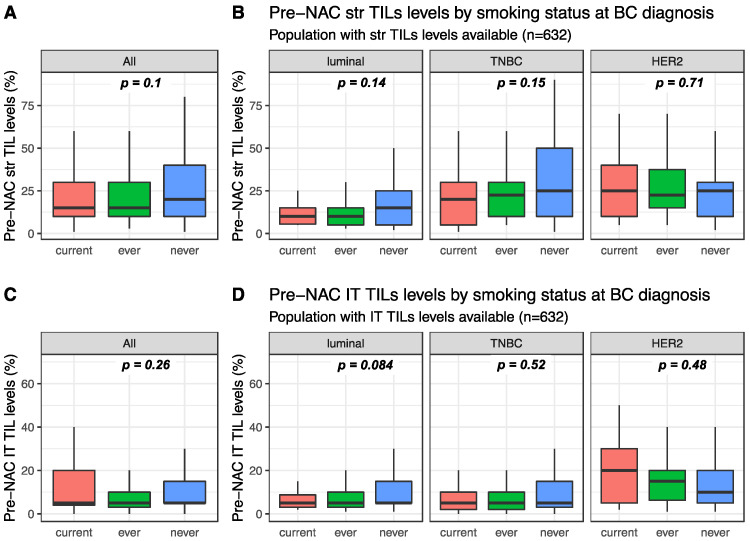
Pre-NAC TIL levels at breast cancer (BC) diagnosis according to smoking status: (**A**) Pre-NAC str TIL levels in the whole population, (**B**) Pre-NAC str TIL levels by BC subtype, (**C**) Pre-NAC IT TIL levels in the whole population, (**D**) Pre-NAC IT TIL levels by BC subtype.

**Figure 2 cancers-12-02943-f002:**
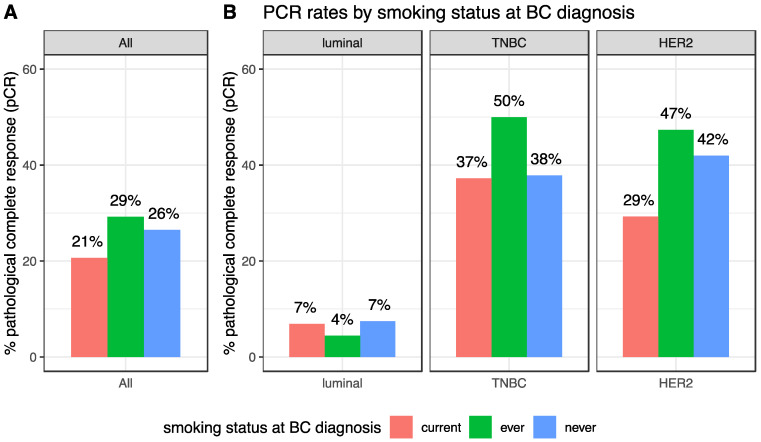
pCR rates according to smoking status at BC diagnosis, in the whole population (**A**) and by BC subtype (**B**).

**Figure 3 cancers-12-02943-f003:**
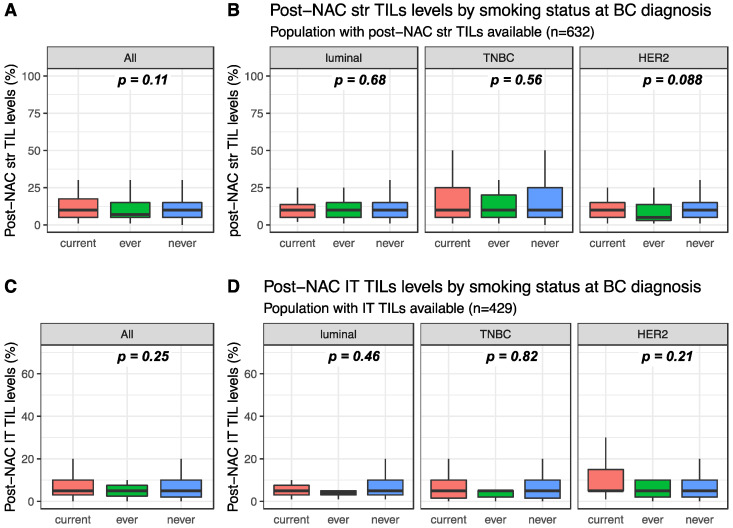
Post-NAC TIL levels according to smoking status: (**A**) post-NAC str TIL levels in the whole population, (**B**) post-NAC str TIL levels by BC subtype, (**C**) post-NAC IT TIL levels in the whole population, (**D**) post-NAC IT TIL levels by BC subtype.

**Figure 4 cancers-12-02943-f004:**
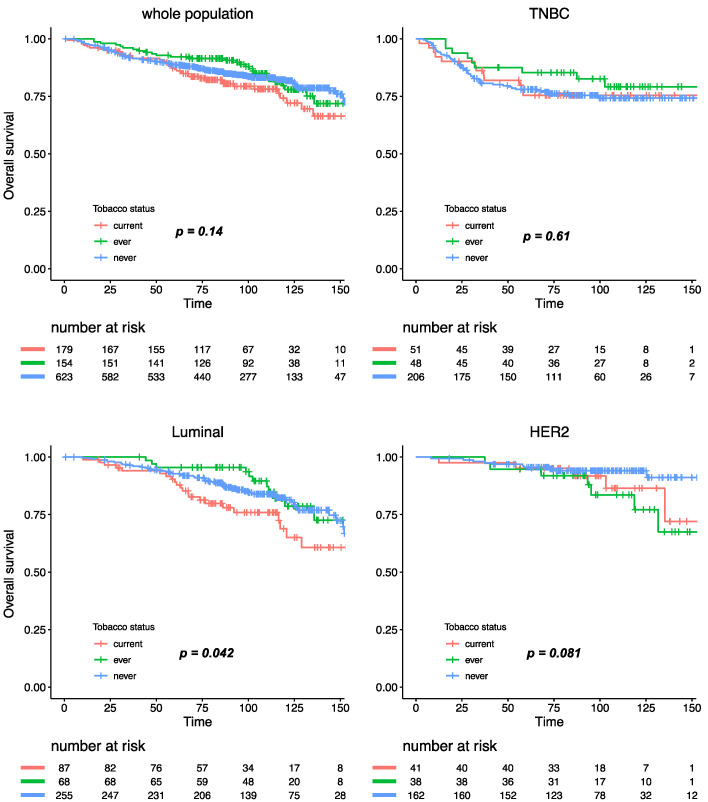
Relapse-free survival according to smoking status in the whole population and by BC subtype.

**Table 1 cancers-12-02943-t001:** Patients’ characteristics among the whole population and according to tobacco status.

Characteristics	Whole Population	Tobacco Status			*p*
		Current	Ever	Never	
	*n* = 956	*n* = 179 (18.7%)	*n* = 154 (16.1%)	*n* = 623 (65.2%)	
	*n* (%)/Mean (SD)/Median (IQR)	*n* (%)/Mean (SD)/Median (IQR)			
Age	47.98 (10.15)	46.5 (8.8)	46.3 (9.7)	48.8 (10.5)	0.002
Menopausal status					0.015
Premenopausal	620 (65.1)	127 (72.2)	108 (70.1)	385 (61.9)	
Postmenopausal	332 (34.9)	49 (27.8)	46 (29.9)	237 (38.1)	
BMI	24.72 (4.65)	24.2 (4.8)	23.9 (4.3)	25.1 (4.7)	0.006
BMI < 19	60 (6.3)	16 (8.9)	12 (7.8)	32 (5.1)	0.007
BMI: 19 to 25	538 (56.3)	106 (59.2)	101 (65.6)	331 (53.1)	
BMI: 25 to 30	233 (24.4)	35 (19.6)	25 (16.2)	173 (27.8)	
BMI > 30	125 (13.1)	22 (12.3)	16 (10.4)	87 (14.0)	
Tumor size (mm)	44.92 (20.40)	44.6 (19.6)	42.1 (18.8)	45.7 (21.0)	0.139
Clinical Tumor Stage					0.251
T1	62 (6.5)	12 (6.7)	12 (7.8)	38 (6.1)	
T2	632 (66.2)	119 (66.5)	111 (72.1)	402 (64.6)	
T3	261 (27.3)	48 (26.8)	31 (20.1)	182 (29.3)	
Clinical nodal status					0.590
N0	405 (42.4)	70 (39.1)	65 (42.2)	270 (43.4)	
N1-N2-N3	550 (57.6)	109 (60.9)	89 (57.8)	352 (56.6)	
Pre-NAC Mitotic index	25.1 (22.0)	24.6 (24.3)	25.5 (22.8)	25.1 (21.1)	0.958
Histology					0.555
NST	854 (90.3)	164 (92.1)	141 (92.2)	549 (89.3)	
Lobular	57 (6.0)	11 (6.2)	8 (5.2)	38 (6.2)	
Medullar	5 (0.5)	1 (0.6)	0 (0.0)	4 (0.7)	
Others	30 (3.2)	2 (1.1)	4 (2.6)	24 (3.9)	
Tumor Grade					0.137
Grade I-II	362 (39.0)	79 (45.7)	57 (38.0)	226 (37.4)	
Grade III	566 (61.0)	94 (54.3)	93 (62.0)	379 (62.6)	
Ki67					
ki67 < 20%	114 (29.2)	26 (32.5)	18 (32.1)	70 (27.5)	0.596
ki67 > 20%	277 (70.8)	54 (67.5)	38 (67.9)	185 (72.5)	
ER status					0.738
ER negative	441 (46.1)	78 (43.6)	71 (46.1)	292 (46.9)	
ER positive	515 (53.9)	101 (56.4)	83 (53.9)	331 (53.1)	
PR status					0.183
PR negative	539 (57.8)	92 (52.3)	84 (56.0)	362 (59.6)	
PR positive	394 (42.2)	84 (47.7)	66 (44.0)	245 (40.4)	
HER2 status					0.692
HER2 negative	715 (74.8)	138 (77.1)	116 (75.3)	462 (74.2)	
HER2 positive	241 (25.2)	41 (22.9)	38 (24.7)	161 (25.8)	
BC Subtype					0.482
luminal	410 (42.9)	87 (48.6)	68 (44.2)	257 (41.3)	
TNBC	305 (31.9)	51 (28.5)	48 (31.2)	205 (32.9)	
HER2	241 (25.2)	41 (22.9)	38 (24.7)	161 (25.8)	
Pre-NAC str TILs *	20 (10, 30)	15.0 (10.0, 30.0)	15.0 (10.0, 30.0)	20.0 (10.0, 40.0)	0.102
Pre-NAC IT TILs *	5 (5, 15)	5.0 (4.0, 20.0)	5.0 (3.0, 10.0)	5.0 (5.0, 15.0)	0.262
NAC regimen					
Anthracyclines-based	100 (10.5)	20 (11.2)	19 (12.3)	61 (9.8)	0.912
Anthracyclines-taxanes	732 (76.6)	136 (76.0)	115 (74.7)	481 (77.2)	
Others	124 (13.0)	23 (12.8)	20 (13.0)	81 (13.0)	

Missing data: menopausal status, *n* = 4; tumor size (mm), *n* = 1; clinical tumor stage, *n* = 1; clinical nodal status, *n* = 1; mitotic index, *n* = 338; tumor Grade, *n* = 28; ki67, *n* = 565; histology, *n* = 10; PR status, *n* = 23; pre-neoadjuvant chemotherapy (NAC) str tumor-infiltrating lymphocytes (TILs), *n* = 324; pre-NAC IT TILs, *n* = 324. Abbreviations: BMI = body mass index; NST = no special type; ER = estrogen receptor; PR = progesterone receptor; TNBC = triple negative breast cancer; NAC = neoadjuvant chemotherapy; str = stromal; IT = intra-tumoral. The “*n*” denotes the number of patients. In the case of categorical variables, percentages are expressed between brackets. In the case of continuous variables, the thmean value is reported, with standard deviation (SD) between brackets. In the case of nonnormal continuous variables *, the median value is reported, with interquartile range between brackets (IQR).

**Table 2 cancers-12-02943-t002:** Post-NAC TILs, pCR and RCB class according to smoking status.

**Characteristics**	**Whole Population**			***p***	**Luminal**			***p***
	**Current** ***n* = 179**	**Tobacco Status**	**Never** ***n* = 623**		**Current** ***n* = 87**	**Tobacco Status**	**Never** ***n* = 255**	
		**Ever**				**Ever**		
		***n* = 154**				***n* = 68**		
	***n* (%)/Mean (SD)/Median (IQR)**				***n* (%)/Mean (SD)/Median (IQR)**			
pCR status				0.169				0.677
No pCR	142 (79.3)	109 (70.8)	458 (73.5)		81 (93.1)	65 (95.6)	236 (92.5)	
pCR	37 (20.7)	45 (29.2)	165 (26.5)		6 (6.9)	3 (4.4)	19 (7.5)	
Post-NAC nodal involvement				0.966				0.942
0	102 (57.0)	94 (61.0)	367 (58.9)		36 (41.4)	24 (35.3)	95 (37.3)	
1 to 3	52 (29.1)	40 (26.0)	172 (27.6)		33 (37.9)	29 (42.6)	107 (42.0)	
≥4	25 (14.0)	20 (13.0)	84 (13.5)		18 (20.7)	15 (22.1)	53 (20.8)	
RCB class	1.9 (1.4)	1.6 (1.4)	1.8 (1.4)	0.44				
pCR	29 (25.2)	33 (30.8)	117 (28.5)	0.706	2 (5.3)	2 (5.1)	7 (5.2)	0.424
1	11 (9.6)	11 (10.3)	39 (9.5)		2 (5.3)	3 (7.7)	13 (9.7)	
2	45 (39.1)	44 (41.1)	177 (43.2)		13 (34.2)	20 (51.3)	68 (50.7)	
3	30 (26.1)	19 (17.8)	77 (18.8)		21 (55.3)	14 (35.9)	46 (34.3)	
Post-NAC LVI				0.883				0.920
no	86 (67.7)	64 (68.8)	294 (66.4)		41 (57.7)	30 (57.7)	126 (60.0)	
yes	41 (32.3)	29 (31.2)	149 (33.6)		30 (42.3)	22 (42.3)	84 (40.0)	
Post-NAC stromal TILs *	10 (5, 17.50)	7 (5, 15)	10 (5, 15)	0.108	10 (5, 13.75)	10 (5, 15)	10 (5, 15)	0.685
Post-NAC IT TILs *	5 (3, 10)	5 (2.50, 7.50)	5 (2, 10)	0.254	5 (3, 7.50)	4 (3, 5)	5 (3, 10)	0.459
**Characteristics**	**TNBC**			***p***	**HER2**			***p***
	**Current** ***n* = 51**	**Tobacco Status**	**Never** ***n* = 206**		**Current** ***n* = 41**	**Tobacco Status**	**Never** ***n* = 162**	
		**Ever**				**Ever**		
		***n* = 48**				***n* = 38**		
	***n* (%)/Mean (SD)/Median (IQR)**				***n* (%)/Mean (SD)/Median (IQR)**			
pCR status				0.280				0.220
No pCR	32 (62.7)	24 (50.0)	128 (62.1)		29 (70.7)	20 (52.6)	94 (58.0)	
pCR	19 (37.3)	24 (50.0)	78 (37.9)		12 (29.3)	18 (47.4)	68 (42.0)	
Post-NAC nodal involvement				0.877				0.629
0	38 (74.5)	39 (81.2)	159 (77.2)		28 (68.3)	31 (81.6)	113 (69.8)	
1 to 3	9 (17.6)	5 (10.4)	28 (13.6)		10 (24.4)	6 (15.8)	37 (22.8)	
≥4	4 (7.8)	4 (8.3)	19 (9.2)		3 (7.3)	1 (2.6)	12 (7.4)	
RCB class								
pCR	17 (37.8)	19 (45.2)	70 (40.5)	0.929	10 (31.2)	12 (46.2)	40 (38.8)	0.885
1	3 (6.7)	4 (9.5)	13 (7.5)		6 (18.8)	4 (15.4)	13 (12.6)	
2	18 (40.0)	16 (38.1)	68 (39.3)		14 (43.8)	8 (30.8)	41 (39.8)	
3	7 (15.6)	3 (7.1)	22 (12.7)		2 (6.2)	2 (7.7)	9 (8.7)	
Post-NAC LVI				0.551				0.228
no	20 (76.9)	17 (85.0)	105 (73.9)		25 (83.3)	17 (81.0)	63 (69.2)	
yes	6 (23.1)	3 (15.0)	37 (26.1)		5 (16.7)	4 (19.0)	28 (30.8)	
Post-NAC stromal TILs *	10 (5, 25)	10 (5, 20)	10 (5, 25)	0.563	10 (5, 15)	5 (3, 13.75)	10 (5, 15)	0.088
Post-NAC IT TILs *	5 (1.50, 10)	5 (2, 5)	5 (1.50, 10)	0.821	5 (5, 15)	5 (2, 10)	5 (2, 10)	0.210

Missing data: RCB class, *n* = 80; Post-NAC LVI, *n* = 99; Post-NAC str TILs, *n* = 80; Post-NAC IT TILS, *n* = 145. Abbreviations: pCR = pathological complete response; NAC = neoadjuvant chemotherapy; RCB = residual cancer burden; LVI = lympho-vascular involvement; str = stromal; IT = intra-tumoral. The “*n*” denotes the number of patients. In the case of categorical variables, percentages are expressed between brackets. In the case of continuous variables, the mean value is reported, with standard deviation (SD) between brackets. In the case of nonnormal continuous variables *, the median value is reported, with interquartile range between brackets.

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
