# Peer review of "No Impact of Smoking Status on Breast Cancer Tumor Infiltrating Lymphocytes, Response to Neoadjuvant Chemotherapy and Prognosis"

_cancers, 2020, doi:10.3390/cancers12102943_

Round 1

Reviewer 1 Report

The manuscript by Simon V et Al. provides intriguing evidence on the lack of impact exerted by smoking on breast cancer infiltrating lymphocytes and clinical outcome following neoadjuvant chemotherapy. Data may be considered controversial within the general believe of smoking associated adverse effects on several aspects of cancer. However, the attempt to dissect the precise role of smoking/nicotine on the composition of tumor immune contexture, and to a more extent on its changes following neoadjuvant treatment, deserves credit.

Here below are my comments/criticisms on the limitations of the study:

  • Heterogeneity is a characteristic feature of cancer also in terms of immune contexture. Do the Authors have any chance to categorize “hot vs cold” tumors, at least based on the abundance of Tumor Infiltrating Lymphocytes (TILs, i.e. cut-off by CART tree analysis), in the whole population and within the different subcategories of breast cancer and then determine whether smoking status impacts on these parameters and clinical outcome?

  • A generally approved methodology was employed here to quantify lymphocytes in tumor samples (ref 46-47). However, Authors should acknowledge that the lack of an immunohistochemical characterization of TILs subtypes strongly limits the actual robustness of the current results. In addition, it should be pointed out that “gross” score evaluations resulting in net numbers (5, 10, 20), instead of assessing the numerical density (n/mm2) of TILs, might not depict the intrinsic variability among and within patient categories, potentially affecting the obtained results.

  • Data widely reported throughout the manuscript indicate that the number of TILs located in stromal areas decreases following neoadjuvant therapy (Post NAC str TILs). Due to the cytotoxicity and potential chemoattractive role of chemotherapy, time lapse from therapy to biopsy may be critical to claim conclusions or provide speculative observations on the effect of treatment on TILs recruitment or reduction. Do you know whether the time interval from NAC to biopsy was similar among subgroups of patients?

  • Although, as properly indicated in the manuscript, the impact of smoking/nicotine differs among tissues, a systemic smoking signature with an inflammatory trait, as repeatedly suggested in NSCLC (i.e. Sato K, Mimaki S, et Al. Lung Cancer. 2020 Jul 1;147:12-20. doi: 10.1016/j.lungcan.2020.06.029.), may be also present in BC patients. Are the Authors investigating this aspect of the ongoing research either by the analysis of circulating phenotypes or DNA by tissue or liquid biopsy?

  • As shown in Table 1, Grade I-II and receptor positive BC were more represented in current smokers compared to the other two groups. Could you comment on the possibility that this variable may be considered in the overall interpretation of the present results?

  • A previous study from your lab (ref 42) documented that TILs located in stromal compartments post neoadjuvant treatment (post NAC s Str TILs) negatively affect survival outcome in patients with HER2 positive breast cancer. Did you validate this finding in the present patient population or is the same cohort of BC? Which are the mechanistic pathways implicated in this phenomenon?

  • I believe that lymphocytes engaged in a close cross talk with tumor cells (IT) may be more clinically relevant than those distally located and to a more extent when dipped in fibrotic tissues therefore featuring immune-exclusion. I noticed a sharp (perhaps not statistically different) increase of pre-NAC IT TILs in HER-2 positive compared to luminal and TNBC types. In addition, current smokers display the highest value of IT TILs in the same group. This effect of smoking seems to disappear following neoadjuvant therapy. Could you please comment or speculate on these findings and their potential underlying mechanisms?

Author Response

We thank reviewer #1 for the essential questions he/she raised and the advices he suggested. We carefully reconsidered the whole manuscript and we applied all changes asked.

Please see the attachment for our point by point response.

Reviewer 2 Report

cancers-915908 Review

No impact of smoking status on breast cancer tumor infiltrating lymphocytes, response to neoadjuvant chemotherapy and prognosis

This study lacks the novelty to deny previous study (J Transl Med 2019, 17:13). To prove the novelty, it is necessary to add basic research on molecular biology. Also, the research design is not clear.

.

Author Response

We thank reviewer #2 for his advice which helped us reconsider the external validity of our study to improve the quality of our manuscript by enriching our discussion.

Please see the attachment for our point by point response.

Reviewer 3 Report

Results:

Why is the pCR after NAC rate in luminal cancer much lower as in the more aggressive TNBC and HER2+ subtypes?

Discussion:

the differences in findings between the authors study and the study from Takada should be discussed more in detail.

The authors should discuss possible explanations for their findings. A critical review of the read-outs and strength and weakness of the different parameters is missing as well.

Minor comments: some formatting issue in table1.

Author Response

We thank reviewer #3 for the time he took to review our manuscript and for the strong added value of his comment. We included his suggestions in our revised manuscript. We believe that our manuscript gained both precision and clarity, and might be more relevant to the reader. 

Please see the attachment for our point by point response.

Round 2

Reviewer 2 Report

This paper has been improved according to the instructions. I think it is acceptable.

Reviewer 3 Report

the authors adressed all questions and comments appropriately and significantly improved the manuscript